# RELLISUR: A Real Low-Light Image Super-Resolution Dataset

**Andreas Aakerberg**[1], **Kamal Nasrollahi**[1,2], **Thomas B. Moeslund**[1]

[1] Visual Analysis and Perception, Aalborg University, Denmark
[2] Research Department, Milestone Systems A/S, Denmark
`anaa,kn,tbm@create.aau.dk`

## Abstract

In this paper, we introduce RELLISUR, a novel dataset of real low-light low-resolution images paired with normal-light high-resolution reference image counterparts. With this dataset, we seek to fill the gap between low-light image enhancement and low-resolution image enhancement (Super-Resolution (SR)) which is currently only being addressed separately in the literature, even though the visibility of real-world images are often limited by both low-light and low-resolution. Part of the reason for this, is the lack of a large-scale dataset. To this end, we release a dataset with 12750 paired images of different resolutions and degrees of low-light illumination, to facilitate learning of deep-learning based models that can perform a direct mapping from degraded images with low visibility to sharp and detail rich images of high resolution. Additionally, we provide a benchmark of the existing methods for separate Low Light Enhancement (LLE) and SR on the proposed dataset along with experiments with joint LLE and SR. The latter shows that joint processing results in more accurate reconstructions with better perceptual quality compared to sequential processing of the images. With this, we confirm that the new RELLISUR dataset can be useful for future machine learning research aimed at solving simultaneous image LLE and SR. The dataset is available at: `https://doi.org/10.5281/zenodo.5234969`.

## 1 Introduction

Digital images can suffer from several different degradations that reduce the visibility and level of details in the images. These degradations can occur both due to environmental factors in the scene, and limitations of the hardware. Two common degradation types are under-exposure, due to poor illumination of the scene, and low resolution, due to the limited spatial resolution of the image sensor. However, with the recent advancements in Convolutional Neural Networks (CNNs), the performance of image processing techniques, such as Low Light Enhancement (LLE) and image Super-Resolution (SR), that can counteract these degradations have been consistently improving.

Imaging in low-light conditions is very challenging due to the low photon count, which leads to low Signal-to-Noise Ratios (SNRs). While increasing the exposure time and ISO setting will result in brighter images, this can also introduce unwanted motion blur and noise. As such, it is difficult to capture high-quality recordings at typical video frame rates in low-light conditions without using external illumination, which is not always a possibility. Simply increasing the brightness of a Low Light (LL) image in postprocessing, will cause the artifacts introduced by the low SNR to be amplified as well. LLE is an active research field that aims to convert degraded LL images to normally exposed high-quality images. However, this is a challenging task as not only the brightness, but also more complex degradations such as color distortion and noise needs to be considered.

35th Conference on Neural Information Processing Systems (NeurIPS 2021) Track on Datasets and Benchmarks.

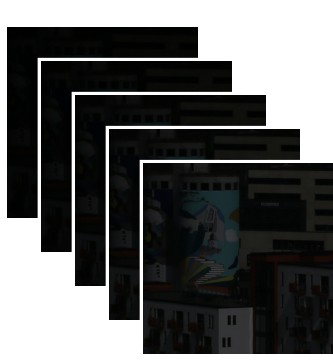 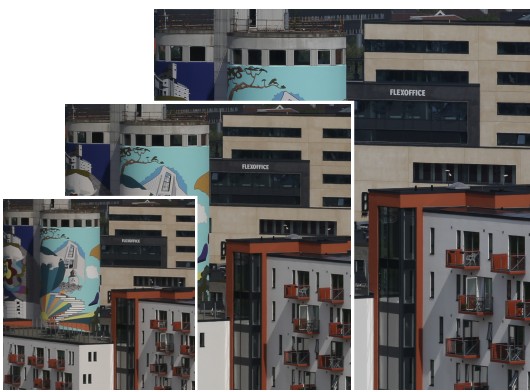

-5.0 to -3.0 EV low light images       Corresponding ×1, ×2, and ×4 normal light images

Figure 1: Example of a sequence of aligned images with different exposure (left) and scale levels (right) from the Real Low-Light Image Super-Resolution (RELLISUR) dataset.

While the resolution of digital cameras has generally increased recently, many cameras are used in combination with lenses with a wide field of view. This leaves very few pixels to resolve objects of interest, such as faces or license plates which can be critical in forensics applications. Hence, it is often desirable to increase the resolution of the Low-Resolution (LR) images to reveal more details. Image SR aims at reconstructing a High-Resolution (HR) image from its LR counterpart. However, most SR methods are trained and evaluated on datasets with synthetically created LR images, where the degradation is assumed to be an ideal bicubic downsampling kernel. This makes these methods unsuitable for real LR images where the degradation models are much more complex [1].

Even though the visibility of real images is often degraded simultaneously by multiple factors, such as low illumination and low resolution, these problems have only been addressed separately in the literature by dedicated LLE and SR methods. However, recent studies have investigated the effect of jointly performing two image processing tasks, e.g. joint LLE and deblurring [2], joint demosaicing and SR [3], and joint denoising and SR [4]. In all of these works, it was found that the joint processing outperforms sequential processing. The is mainly due to the accumulation of errors produced by the individual methods, and the possibility of early algorithms removing information that could be valuable for subsequent processing. We believe that part of the reason for why joint LLE and SR of real images has not yet been investigated in the literature, is due to the lack of a large-scale dataset of paired Low Light Low Resolution (LLLR) and Normal Light High Resolution (NLHR) images. Hence, we argue that such a dataset is of major importance in the image processing, computer vision, and machine learning community with the advent of deep-learning based methods which performance is highly dependant on data [5]. Furthermore, it is highly desired that such a dataset consists of real-world LLLR and NLHR image pairs, as opposed to synthetic image pairs, in order to allow algorithms to generalize to practical applications. However, constructing such a dataset is a non-trivial task as real image pairs are difficult to obtain.

In this work, we present the **RE**al **L**ow-**L**ight **I**mage **SU**per-**R**esolution (RELLISUR) dataset which is the first dataset to contain real LLLR and NLHR image pairs. The dataset is made publicly available and contains a large number of in- and outdoor scenes captured by a Digital single-lens reflex (DSLR) camera. There are more than 12000 image-pairs of diverse content and degradation strength in the dataset, which is more than sufficient to train Deep neural networks (DNNs). Applications of the dataset include remote sensing, surveillance, and forensics among others. Figure 1 shows an example of a sequence from RELLISUR containing aligned LLLR and NLHR images of the same scene.

Our collection method is reproducible and easy to follow. We collect images of different resolutions from the same static scene by changing the focal length of a zoom lens. An increasing amount of details are obtained as the focal length is increased. Along with images of different resolution, we also collect corresponding images of different low-light levels. We obtain the low-light images by shortening the exposure time. As the changing focal lengths naturally introduce misalignment between the image pairs, mainly due to varying lens distortion, we develop an effective post processing pipeline to align the image pairs.

LLE or SR are both ill-posed problems, and as such, simultaneously reconstructing images degraded

by both LL and LR images is a highly challenging problem. To analyze the effectiveness of RELLISUR in this regard, we train and evaluate both dedicated models for each task as well as models for joint LLE and SR. The experimental results demonstrate the value of RELLISUR by showing that joint processing outperforms sequential processing. Thus, we hope that the RELLISUR can help facilitate further work in joint LLE and SR.

The contributions of our work are summarized as follows:

- We present the first large-scale dataset of paired and aligned low-light/low-resolution and normal-light/high-resolution images of diverse content, which closes the gap between the LLE and SR problems.
- We provide a comprehensive benchmark of existing methods for separate image LLE and SR along with experiments on joint processing on the proposed dataset.
- We show that joint image LLE and SR leads to better results than sequential processing, which highlights the need for new machine learning methods to handle the LLLR image enhancement problem.

## 2 Related Work

### 2.1 Real-world super-resolution datasets

There exist several image datasets to facilitate training and evaluation of SR methods. These include Set5 [6], Set14 [7], BSD100 [8], and DIV2K [9] among others. However, these datasets only contain the HR image, and the corresponding LR image then has to be created synthetically. The traditional way of doing this is to downsample the HR image with bicubic interpolation. As the real-world image degradation is much more complicated, SR models trained on such data often show poor performance on real LR images due to the domain difference [10, 1]. To overcome this issue, some researchers recently started to collect real LR/HR image pairs. And overview of such datasets can be seen in Table 1. Qu et al. used a beam splitter and two cameras to collect 31 paired LR/HR face images in an indoor lab environment [11]. The City100 dataset by Chen et al. [12] consists of 100 paired images of postcards with cityscapes, captured by DSLR and smartphone cameras. Kohler et al. relied on hardware binning to capture image-pairs of different resolution [1]. The dataset contains 5670 HR images, but the variance and application to real-world scenarios are limited as the dataset only depicts 14 different indoor lab scenes acquired in grayscale. Zhang et al. collected 500 scenes of LR/HR resolution using a DSLR camera equipped with a zoom lens, which made it possible to obtain images with varying degrees of detail [13]. Images captured with a long focal length contain finer details compared to an image of the same scene captured with a short focal length. However, the images in this dataset are not pixel-wise aligned, which complicates the learning of a mapping from LR to HR. Cai et al. [14] proposed an image registration algorithm to align 243 LR/HR pairs collected with two DSLR cameras and using different focal lengths of a zoom lens. The images in the dataset depict various outdoor scenes and objects located indoors. However, a limitation of this dataset is the number of images, as there are only 175 pairs for the $\times 4$ scale. Most recently Wei et al. proposed the DRealSR dataset [15] which contains a total of 2507 LR/HR image pairs collected with five different cameras using different zoom-lens focal lengths.

However, as all of the existing real SR dataset contains image pairs where the illumination of the HR images is consistent with that of the LR images, SR models trained on such data naturally perform poorly on low-light images.

### 2.2 Low/normal-light datasets

Only very few datasets of paired low/normal-light images captured in real scenes exist. The LOL dataset [16] contains 500 low/normal-light image pairs which are all downscaled to a resolution of $600 \times 400$ pixels. The images are captured both in and outdoors at daylight, and the low-light images are created by changing the ISO and exposure settings of the camera, which results in LL images with low contrast, color distortion, and sensor noise due to the low SNR. Unfortunately, the downscaling of the images reduces the natural sensor noise and changes other real-world characteristics [17], such that the images can no longer be considered real LL images. The SID dataset [18] contains 5094 short exposure, and 424 long exposure RAW image pairs of either 12 and 24 MPIX resolution. All images are captured outside at nighttime or indoors in rooms with low illumination. The normal-light images

Table 1: Overview of real-world super-resolution datasets of paired real LR and HR images.

| Name | Year | LR/HR Pairs | Type | HR resolution | Method | Content |
|------|------|-------------|------|---------------|--------|---------|
| Qu et al. [11] | 2016 | 31 | RAW | 2.3MPIX | Beam-splitter | Faces |
| RealSR [14] | 2019 | 595 | RGB | 0.48 to 5.28MPIX | Zoom lens | In/outdoor scenes |
| City100 [12] | 2019 | 100 | RGB | 1.06MPIX | Zoom + translation | Postcards |
| SupER [1] | 2019 | 5,670 | Grayscale | 2.2MPIX | Hardware binning | Indoor lab |
| SR-RAW [13] | 2019 | 500 | RAW | 12MPIX | Zoom lens | In/outdoor scenes |
| DRealSR [15] | 2020 | 2,507 | RGB | 20 to 24MPIX | Zoom lens | In/outdoor scenes |
| Ours | 2021 | 2,250 | RGB | 0.39 to 6.25MPIX | Zoom lens | In/outdoor scenes |

are created by capturing long exposure images of the same static scenes. However, this method leads to Normal Light (NL) images with less vibrant colors than actual daylight images and the risk of locally overexposed areas and excessive noise. In [19] a collection of HDR images along with their SDR counterparts are presented. The HDR sequence contains both under- and over-exposed images. All the existing LLE datasets contain LL and NL image pairs of the same spatial resolution, which means that they are not feasible to use for jointly handling the LLE and SR problem. An overview of the datasets can be seen in Table 2.

Table 2: Overview of low-light image datasets with LL and NL pairs.

| Name | Year | GT images | LL/NL Pairs | Type | Resolution | Method |
|------|------|-----------|-------------|------|------------|--------|
| LOL [16] | 2018 | 500 | 500 | RGB | 0.24MPIX | Normal + under-exposure |
| SID [18] | 2018 | 424 | 5,094 | RAW | 12/24MPIX | Under + long-exposure |
| SICE [19] | 2018 | 589 | 4,413 | RGB | 6 to 24MPIX | HDR |
| Ours | 2021 | 2,250 | 12,750 | RGB | 0.39 to 6.25MPIX | Normal + under-exposure |

## 3 RELLISUR Dataset

This section introduces the RELLISUR dataset. We discuss in detail the data collection process, preprocessing, statistics, and present a suggested train/validation/test split.

### 3.1 Collection method

The RELLISUR dataset is a novel collection of image sequences containing real $\times 1$, $\times 2$, and $\times 4$ NL images, together with five real LL images. The $\times 1$ and $\times 2$ scale levels represent the LR images while the $\times 4$ scale level represents the high-resolution Ground-Truth (GT) reference images. The LL images are acquired at scale $\times 1$ and are also considered low-resolution.

The dataset is collected with a Canon EOS 6D camera equipped with a Canon 70-300mm L IS USM zoom lens. Since the size of an object depicted on the image sensor is approximately linear to the focal length [20], a doubling of the scale level can be obtained by doubling the focal length. Hence, to capture images of different scale levels, we used a focal length of 70mm, 140mm, and 280mm to capture the $\times 1$, $\times 2$ and $\times 4$ scale levels, respectively.

All normal light images are captured using auto-exposure, auto-white-balance, and auto-focus using the center focus point only. The exposure metering is set to partial metering. The ISO value is set between 100 and 400 to ensure low noise levels in the NL images. To avoid misalignment issues, we aim at capturing static scenes and minimize camera movement due to wind, which is essential when using a telephoto lens. To minimize camera shake, the camera is mounted on a sturdy tripod, and hence the lens stabilization feature is disabled. To obtain a high depth-of-field we use an f-stop setting of f/22. The camera is triggered remotely to avoid movement.

In photography, the Exposure value (EV) is defined as $log_2 \frac{N^2}{t}$, where $N$ and $t$ are the camera lens f-stop number and exposure time in seconds, respectively. Hence, a decrease of -1.0 EV corresponds to half as long exposure time, or one-stop, in our case as the f-stop is kept fixed. To capture LLLR images with different degrees of under-exposure, we used the camera's auto bracketing mode to

obtain five successive images that are under-exposed in different levels from the auto exposure setting. We used two different ranges going from from -4.5 to -2.5 and -5.0 to -3.0 EV steps. The resulting average exposure times for both the in- and outdoor scenes can be seen in Table 3. This wide range of under exposure levels can help to improve the generalization abilities of models trained on the RELLISUR dataset.

The images in the dataset are collected in natural scenes, both in- and outdoors, and depict architecture, signs, plants, common office items, art, etc. The number of in- and outdoor scenes are nearly identical with a $49\%$ and $51\%$ distribution, respectively. We decided not to collect images that could enable identification of individuals, by avoiding faces, persons, license plates, or other personally identifiable information. Likewise, we avoided capturing images with content that could be considered offensive, insulting, or threatening. We have manually screened the dataset to ensure that all images apply to these requirements.

In total, the RELLISUR dataset consists of 850 distinct sequences. An example of a sequence can be seen in Figure 1. With three different scale levels, the total number of normal light LR and HR pairs is 2550. As the five under-exposed images in a sequence corresponds to the same NL reference image, the resulting number of LL / NL image pairs is 4250 for each of the three scale levels. Hence, the total number of LL / NL images pairs in the RELLISUR is 12750.

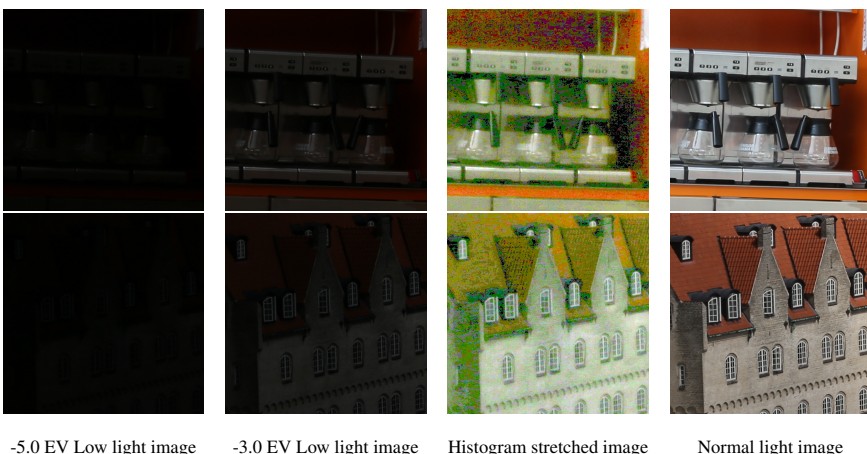

-5.0 EV Low light image  -3.0 EV Low light image  Histogram stretched image  Normal light image

Figure 2: Examples of the noise and color distortion in the under-exposed images in RELLISUR. To aid visualization, the -5.0 EV LL images have been histogram stretched to match the NL images

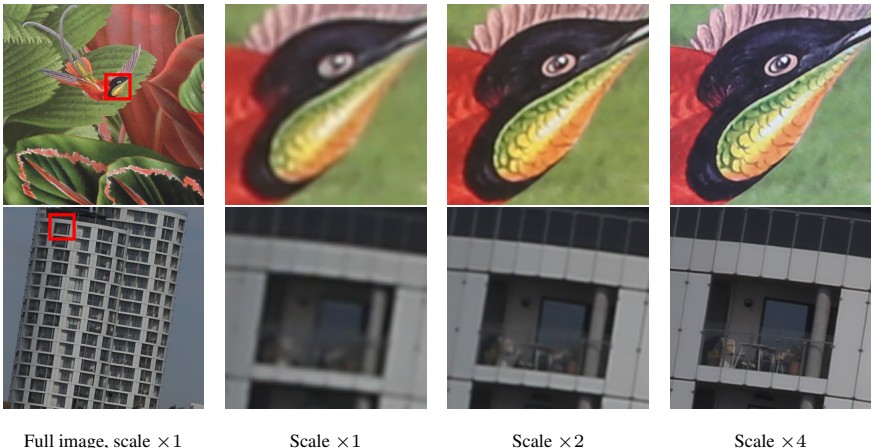

Full image, scale ×1  Scale ×1  Scale ×2  Scale ×4

Figure 3: Examples of the difference in image quality between the scale levels in RELLISUR. To aid visualization we show image crops.

## 3.2 Preprocessing

During the collection of image sequences, multiple factors can unintentionally affect the images quality negatively. First, the lens characteristics change when zooming, resulting in different levels of warping and distortion of the image. Next, external factors, such as wind, can affect the camera causing a slight shift in the scene or motion blur. To mitigate this, we apply a carefully designed preprocessing scheme to the collected images.

First, we manually screen the collected sequences and discard ones that contain images which are out-of-focus, incorrectly exposed, contain moving objects, or other undesired defects. Next, we apply lens correction in Adobe Lightroom [21] using the appropriate lens and camera profiles. This removes chromatic aberration and corrects the lens distortion. However, as the corner regions of the images are difficult to undistort, and also less sharp than the center part, we center crop the $\times 4$ NLHR reference images to the center $2500 \times 2500$ pixels. Although the images are now distortion-free, the individual images in a sequence are not guaranteed to be pixel-wise aligned due to inability to accurately adjust the zoom lens at the exact desired focal lengths. Furthermore, the optical center of the lens might shift slightly during zooming [22]. To register all images in a sequence to match the $\times 4$ NLHR reference image, we first detect and match SURF [23] features between the $\times 1$ and $\times 4$ NL images for a given sequence. To maintain the spatial resolution difference of the three scale levels we use a downsampled version of the $\times 4$ NLHR as target. Then, we use the matched coordinates to estimate a homography using MSAC [24]. Using the translation parameters, we crop and align both the $\times 1$ LL and NL images to the $\times 4$ NLHR reference image. Lastly, we use the same method to register the $\times 2$ NL image to the $\times 4$ NLHR reference image. As such, the resolution of the $\times 1$ and $\times 2$ images become $625 \times 625$ and $1250 \times 1250$ pixels, respectively. An overview of the preprocessing pipeline can be seen in Figure 4. One limitation of RELLISUR is that the LL images are so dark that it is impossible to verify if something undesired has entered the scene, such as a bird flying by. Furthermore, changes in environmental lighting conditions can affect the brightness of the images within a sequence. Considering that this does not affect a model's ability to learn to solve the LLE problem, we do not attempt to match the brightness levels.

Lastly, we partition the dataset into train, validation, and test splits, with a 85%/5%/10% distribution, respectively. This results in 722 train, 43 validation, and 85 test sequences. We encourage researchers to use this split to enable direct comparison with future works.

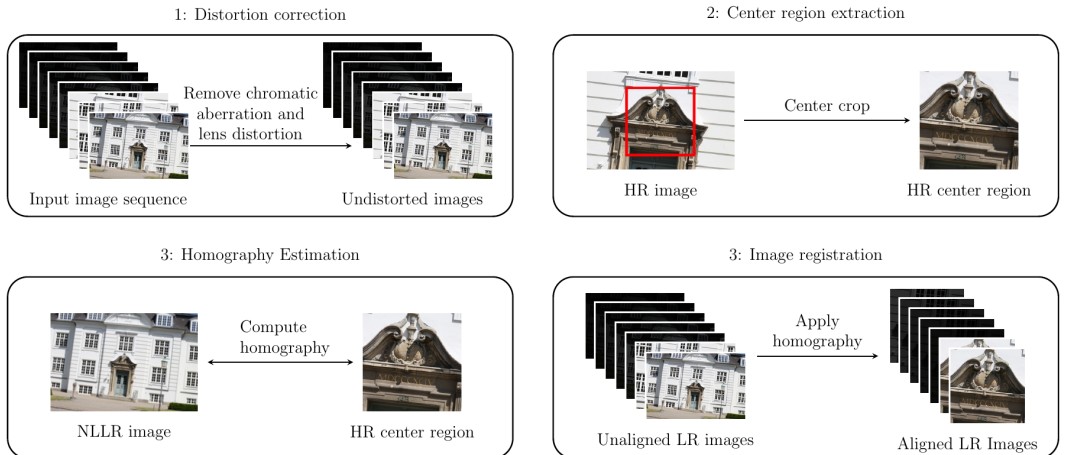

Figure 4: Overview of the preprocessing pipeline.

## 3.3 Analysis of dataset content

As seen in Figure 3, the three different scale levels of the NL images in RELLISUR are all properly exposed and noise free, but have a clear difference in details and sharpness. In comparison, the LL images lacks contrast and contains strong color distortion and sensor noise, as illustrated in Figure 2. The average pixel value of the LL and NL images is shown in Figure 5. Here it can be seen that most of the pixel values of the LL images are below 50, while the ones of the NL image are more evenly distributed across the range. This is supported by the average mean $\mu$ and standard deviation $\sigma$ values

computed on grayscaled versions of the images in the dataset. As seen in Table 4, the average pixel values of the LL images in RELLISUR are lower and less spread compared to the ones in the widely used LOL dataset [16], which indicates that the LLE task on RELLISUR is more challenging. To quantify how the different levels of under-exposure degrades the image quality, we have computed the average Peak Signal-to-Noise Ratio (PSNR), Structural Similarity index (SSIM) [25] and LPIPS [26] quality scores for each of the different EV ranges against the properly exposed images. As seen in Table 3, both the fidelity and perceptual quality drops significantly as the exposure time is decreased. Furthermore, the average exposure times for indoor scenes are longer than the ones for outdoors scenes, mainly due to differences in available light.

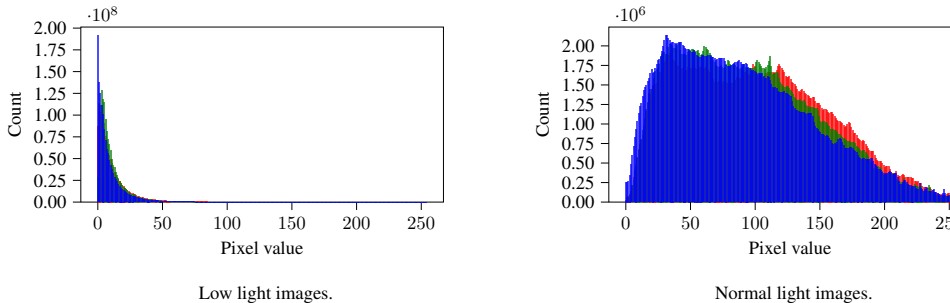

Low light images.

Normal light images.

Figure 5: Average RGB histograms of the low light and normal light reference images in RELLISUR. The horizontal axis represents the pixel value and the vertical axis the number of occurrences.

Table 3: Average decrease in image quality and exposure time for the different under-exposure levels in RELLISUR

| Exposure | PSNR ↑ | SSIM ↑ | LPIPS↓ | Indoor | Outdoor |
|---|---|---|---|---|---|
| Auto | $\infty$ | $\infty$ | $\infty$ | 1.722s | 0.095s |
| -2.5 EV | 10.35 | 0.30 | 0.46 | 0.505s | 0.025s |
| -3.0 EV | 9.40 | 0.22 | 0.57 | 0.212s | 0.013s |
| -3.5 EV | 8.82 | 0.16 | 0.67 | 0.152s | 0.009s |
| -4.0 EV | 8.42 | 0.11 | 0.76 | 0.107s | 0.006s |
| -4.5 EV | 8.13 | 0.07 | 0.84 | 0.077s | 0.004s |
| -5.0 EV | 7.87 | 0.05 | 0.89 | 0.031s | 0.003s |

Table 4: Average mean $\mu$ and standard deviation $\sigma$ values.

| Name | LOL [16] | Ours |
|---|---|---|
| $\mu$ LL | 15.48 | 10.59 |
| $\sigma$ LL | 10.40 | 8.14 |
| $\mu$ NL | 116.92 | 96.35 |
| $\sigma$ NL | 45.96 | 47.73 |

## 4 Experiments

We conduct several experiments on the RELLISUR dataset to evaluate its usefulness for future research on the development of machine learning models for end-to-end mapping from LLLR to NLHR. All experiments are done using the splits defined in section 3.2.

Since no publicly available methods for joint LLE and SR of real images currently exist, we first separately benchmark ten different State-of-the-Art (SoTA) LLE and SR methods by training and evaluating them on the RELLISUR dataset. Next, we select the best performing LLE and SR methods, in terms of reconstruction accuracy and perceptual quality, and combine these to sequentially process the LLLR images to obtain NLHR images. Lastly, to verify that the dataset can also be used to learn an end-to-end mapping from LLLR to NLHR, we train an SR model and an LLE model with an added upscaling module. All experiments involving SR are conducted on both scale levels in the dataset ($\times 2$ and $\times 4$).

### 4.1 Baseline methods for end-to-end learning

While SR models are not aimed at enhancing LL images, the ESRGAN [27] is a very capable model with more than 16 million parameters. Furthermore, this model produces HR reconstructions with the best perceptual quality of all the evaluated methods. Hence, we chose this SR model to learn the full end-to-end mapping directly from LLLR to NLHR. As LLE methods are not capable of

increasing the resolution of the input images, these have to be modified in order to be able to learn the end-to-end mapping. For this we choose the MIRNet model it has the LLE performance in terms of reconstruction accuracy. To enable the MIRNet to transform LR to HR images, we add the learnable upsampling module from [28] to the end of the model. This module utilizes sub-pixel convolution [29] for efficient upsampling.

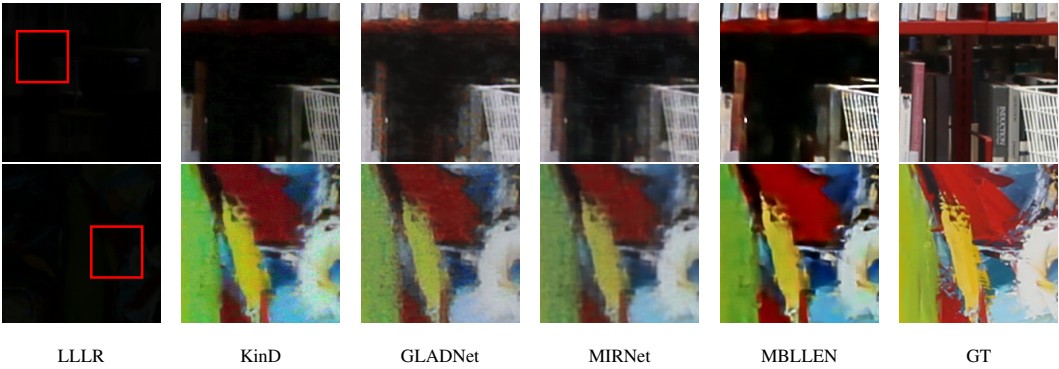

| LLLR | KinD | GLADNet | MIRNet | MBLLEN | GT |

Figure 6: LLE results on the RELLISUR test set by different methods trained on the training set.

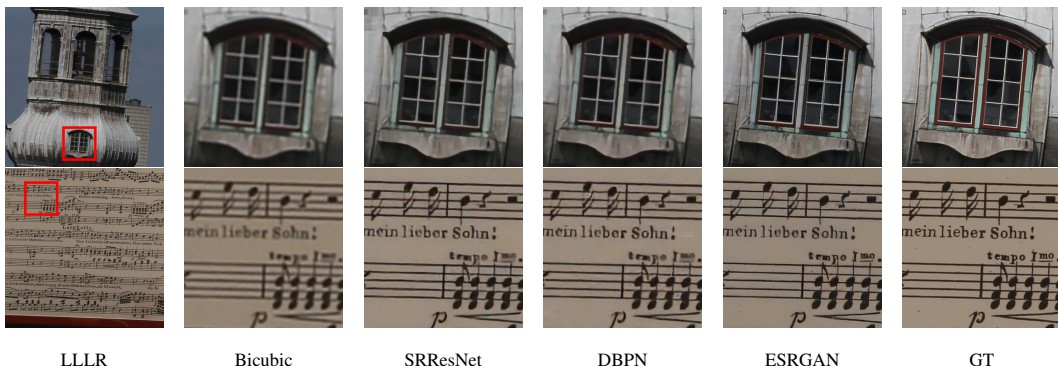

| LLLR | Bicubic | SRResNet | DBPN | ESRGAN | GT |

Figure 7: SR results ($\times 4$) on the RELLISUR test set by different methods trained on the training set.

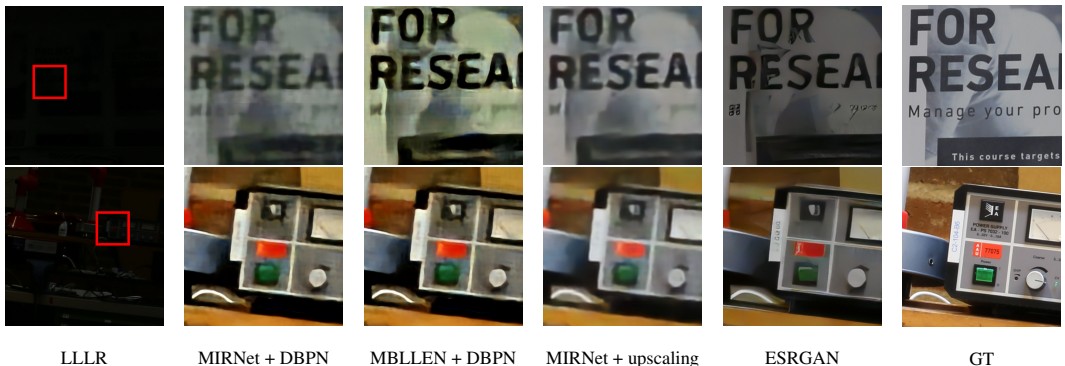

| LLLR | MIRNet + DBPN | MBLLEN + DBPN | MIRNet + upscaling | ESRGAN | GT |

Figure 8: Simultaneous LLE and SR results on the RELLISUR test set by different methods trained on the training set.

## 4.2 Implementation details

All supervised models have been re-trained using the hyperparameter settings described by the authors. The modified MIRNet model was trained for 1000 epochs. We used a single NVIDIA V100 card to perform the training. To evaluate the reconstruction accuracy of the different methods, we

first crop 4 border pixels to avoid boundary artifacts, and calculate the average PSNR and SSIM [25] values on the test set using MATLAB [30]. While these metrics are typically used in LLE and SR research to measure the similarity to GT images, the resulting scores often correlate poorly with perceived similarity. To this end, we also include the more recent LPIPS [26] metric which has shown to correlate better with human judgment. We use the LPIPS implementation provided by the authors and used the weights from the pre-trained AlexNet [31] for evaluation.

## 4.3 Results

Table 5: LLE results for different methods trained and tested on the RELLISUR dataset.

| Name | PSNR ↑ | SSIM ↑ | LPIPS↓ |
|---|---|---|---|
| Zero-DCE [32] | 12.99 | 0.44 | 0.79 |
| Retinex-Net [16] | 15.43 | 0.34 | 0.68 |
| LECARM [33] | 10.04 | 0.25 | 0.53 |
| RUAS [34] | 11.92 | 0.34 | 0.51 |
| LIME [35] | 14.95 | 0.45 | 0.42 |
| EnlightenGAN [36] | 11.61 | 0.39 | 0.39 |
| KinD [37] | 15.84 | 0.49 | 0.33 |
| GLADNet [38] | 21.09 | 0.69 | 0.30 |
| MIRNet [39] | **21.62** | **0.77** | 0.28 |
| MBLLEN [40] | 17.52 | 0.60 | **0.23** |

Table 6: SR results for different methods trained and tested on the RELLISUR dataset.

| Name | ×2 | | | ×4 | | |
|---|---|---|---|---|---|---|
| | PSNR ↑ | SSIM ↑ | LPIPS↓ | PSNR ↑ | SSIM ↑ | LPIPS ↓ |
| Bicubic | 28.70 | 0.91 | 0.20 | 23.97 | 0.82 | 0.43 |
| SRCNN [41] | 29.92 | 0.92 | 0.16 | 24.90 | 0.83 | 0.35 |
| SRFBN [42] | 29.78 | 0.92 | 0.16 | 24.77 | 0.84 | 0.33 |
| RDN [43] | 28.48 | 0.92 | 0.17 | 22.96 | 0.84 | 0.33 |
| SRResNet [44] | 29.82 | 0.92 | 0.15 | 24.52 | 0.84 | 0.32 |
| EDSR [45] | 29.69 | 0.92 | 0.16 | 24.06 | **0.85** | 0.32 |
| DBPN [46] | **29.99** | **0.92** | 0.15 | **24.98** | 0.84 | 0.30 |
| Real-ESRGAN [47] | 27.73 | 0.89 | 0.16 | 23.14 | 0.80 | 0.29 |
| SRGAN [44] | 29.42 | 0.90 | 0.11 | 24.29 | 0.80 | 0.22 |
| ESRGAN [27] | 29.79 | 0.91 | **0.10** | 24.71 | 0.80 | **0.21** |

Table 7: Simultaneous LLE and SR results for different approaches trained and tested on the RELLISUR dataset.

| Type | Name | ×2 | | | ×4 | | |
|---|---|---|---|---|---|---|---|
| | | PSNR ↑ | SSIM ↑ | LPIPS↓ | PSNR ↑ | SSIM ↑ | LPIPS ↓ |
| Sequential | MIRNet + DBPN [39, 46] | 20.73 | 0.73 | 0.49 | 19.85 | 0.74 | 0.58 |
| | MIRNet + ESRGAN [39, 27] | 20.67 | 0.72 | 0.47 | 19.81 | 0.71 | 0.56 |
| | MBLLEN + DBPN [40, 46] | 17.89 | 0.60 | 0.38 | 17.15 | 0.58 | 0.50 |
| | MBLLEN + ESRGAN [40, 27] | 17.74 | 0.56 | 0.40 | 17.03 | 0.50 | 0.52 |
| Joint | MIRNet [39] + Upscaling module | **21.33** | **0.75** | 0.41 | **20.62** | **0.75** | 0.53 |
| | ESRGAN [27] | 17.67 | 0.68 | **0.35** | 17.28 | 0.66 | **0.39** |

As seen in Table 5 the best performing LLE method, according to the hand-crafted PSNR and SSIM metrics, is the MIRNet [39], while the method resulting in the best perceptual quality according to LPIPS [26] is the MBLLEN [40]. A visual comparison can be seen in Figure 6. For the SR methods, as seen in Table 6, the best performing models are the DBPN [46] and ESRGAN [27] in terms of fidelity and perceptual quality, respectively. A visual comparison can be seen in Figure 7.

Regarding simultaneous LLE and SR, we see that sequential processing with the respectively best performing methods, in terms of either PSNR and LPIPS is worse than joint processing. Interestingly, the LLLR images reconstructed with the ESRGAN [27] have the best perceptual quality even though this model is not designed for LLE. At the same time the ESRGAN results in the lowest PSNR value, but this is expected due to the perception distortion tradeoff [48], since this model is optimized to produce visually pleasing images. Conversely, the MIRNet [39] model with the added upscaling module and optimized for low distortion with Charbonnier loss [49], results in the best PSNR and SSIM values. The qualitative results and examples of reconstructed images can be seen in Table 7 and Figure 8, respectively.

## 5  Conclusion

We have argued for the need for a dataset to fill the gap between LLE and SR. To this end, we have introduced the RELLISUR dataset to the community, a novel large-scale collection of paired LLLR and NLHR reference images. We offer the dataset as free and open-source with the purpose of advancing machine learning applications in the area of image processing. We also provided an extensive benchmark of the existing methods for LLE and SR, and highlighted the need for new methods to reconstruct images that are degraded by both low light and low resolution. Additionally, we have experimentally demonstrated that this dataset can be used to train deep-learning-based methods for joint LLE and SR, that outperform sequential processing. As such, we believe the RELLISUR dataset will be valuable for the community.

**Broader impact**   As this dataset contains image data that can be used to improve the performance of LLE and SR algorithms, there is a risk that malicious parties could harness this to develop more capable surveillance systems for monitoring and tracking of people. However, we have carefully screened the dataset to remove any personal information, such as persons and faces, which greatly reduce the possible negative uses of the data. On the positive side, our dataset enables reproducible research on image restoration problems which will aid in advancing these by consistent and reliable baselines.

**Disclosure of Funding**   This research was funded by Milestone Systems A/S, Brøndby Denmark and the Independent Research Fund Denmark, under grant number 8022-00360B.

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
