# OpenReview forum: "RELLISUR: A Real Low-Light Image Super-Resolution Dataset"
_NeurIPS.cc/2021/Track/Datasets_and_Benchmarks/Round2 — NeurIPS 2021 Datasets and Benchmarks Track (Round 2)_

### Official Review · Reviewer_vZ4H · 2021-09-17
**A Real Low-Light Image Super-Resolution Dataset**

**Rating:** 5
**Confidence:** 3
**Correctness:** Yes
**Clarity:** Yes

**Strengths:**

- In general, this paper is well organized and easy to follow.
- The RELLISUR is created with realistic data and has a large size that is comparable to the existing dataset.
- Recent SOTA models are evaluated on the proposed dataset for benchmarking, which is beneficial for future research.


**Weaknesses:**

- Simply combining low-light enhancement (LLE) with super-resolution (SR) forms a very specific task and looks incremental to me. This paper fails to show/analysis the inherent relevance between those two tasks but just combining them together, hence I am concerned about its research value and interest for the community.

- All the samples are collected from static scenes and ignore dynamic content that is also very important in practice.

- Fig.5~7 should choose the same examples so as to support the authors' claim that joint task is better.

- The results in Tab.7 may not support the authors' claim that joint task is better. Even though under the Joint setting MIRNet achieves better performance, the ESRGAN achieves worse results.

- A minor concern is that RELLISUR formulates a low-level computer vision problem more than a machine learning problem. It may be better suitable for some other venues rather than NeurIPS.


**Additional Feedback:**

No

**Documentation:**

Yes

**Relation To Prior Work:**

Yes

**Summary And Contributions:**

This paper presents the RELLISUR, a real dataset for the low-light image super-resolution task. The dataset contains 850 distinct LLLR/NLHR  sequences that are collected by controlling the camera lens and exposure time. Various SOTA models for LLE and SR are applied on RELLISUR for benchmarking.

---

> ### Author Response · Authors · 2021-09-26
> **Response to reviewer vZ4H**
>
> We thank the reviewer for their review. Please find our answers below:
>
> Relation between low-light enhancement and super-resolution:
> Low-light enhancement (LLE) and super-resolution (SR) of real-world images are two highly ill-posed problems, that are difficult to solve on their own, making simultaneous LLE and SR a very challenging research problem. Furthermore, since the visibility of real-world images are often limited by both low-light and low-resolution, there are also many practical applications of simultaneous low-light enhancement and super-resolution e.g. in remote sensing, surveillance, and other scenarios where the illumination of the scene and distance to objects of interest cannot be controlled. In our paper, we show that naively solving the LLE and SR problem with sequential processing leads to sub-optimal results compared to joint processing with methods trained by supervised-learning, which is made possible for the first time by our proposed dataset. In addition, we argue that any future work to improve on the joint LLE and SR problem will most likely also be valuable for a number of other research domains. As such, we believe that our dataset is a valuable contribution to the community.
>
>
> Dynamic content:
> While we agree on the relevance of dynamic content in datasets, simultaneous LLE and SR of static scenes are already a very challenging research problem. Thus, we will leave the introduction of dynamic content for future work.
>
> Quantitative results:
> For our experiments with joint processing, we chose to include two methods, each optimized for their own specific goal, i.e. low reconstruction error or high perceptual quality. Since these goals are at odds with each other, meaning that a low reconstruction error does not necessarily result in visually pleasing images [1], we aim to show that both goals can be separately obtained with our dataset. To this end, we jointly train the MIRNet + upscaling, optimized with Charbonnier loss (which is highly correlated with PSNR), and the ESRGAN which is optimized towards perceptual quality with a combination of L1, perceptual and GAN loss. In our experiments, we measure the reconstruction error with PSNR and the perceptual quality with LPIPS and find that both jointly trained methods outperform sequential processing in terms of either reconstruction error or perceptual quality. To aid understanding of this difference, we have elaborated on this in the results section of our revised manuscript.
>
> Fit:
> Since almost all recent computer vision research is based on machine learning which are often heavily dependent on data, we believe that our dataset is a suitable fit for the NeurIPS dataset track.
>
> [1] Y. Blau, T. Michaeli, "The perception-Distortion Tradeoff", CVPR 2018

---

### Official Review · Reviewer_URub · 2021-09-20
**Great for end-to-end training from low-light low-resolution images to normal-light high-resolution images**

**Rating:** 7
**Confidence:** 5
**Clarity:** The paper is very well written, well …

**Strengths:**

The RELLISUR is the first LL/LR to NL/HR dataset that allows for joint SR and LLE tasks. This dataset contains real-world LL/LR and NL/HR image pairs, as opposed to synthetic image pairs; therefore, it allows SR/LLE algorithms to generalize to practical applications.

One unique advantage of this dataset is that the authors effectively process the image pairs with different resolution scales to allow pixel-wise alignment, which is a challenging task and a remarkable achievement.

As the first of its kind, this dataset will be very useful to researchers aiming at directly inferring NL/HR images from LL/LR ones.

After I download the dataset and examine its content, I found the images were all well marked and separated into the train, validation, and test sets. The images are all png files and can be easily used by most algorithms.

As stated by the authors, the dataset does not contain any personal information; therefore, it should not cause negative consequences as a result of using this dataset.

**Weaknesses:**

The name of this dataset, i.e., RELLISUR, is a little bit confusing. The authors did not provide the full name of the abbreviation and the name sounds strange.

When capturing the images, the authors used a lot of "auto" settings, including auto-exposure, auto-white-balance, and auto-focusing. Due to the unpredictability of these settings, the acquired images might have significantly different exposure, white balance, and focus. These unpredictable imaging properties might have unforeseeable consequences when using these images to train a network.

According to the paper, there should be 850 distinct sequences, and therefore 850x3 = 2550 LR/HR image pairs. However, the number becomes 2250 in the paper.

The preprocessing pipeline in Section 3.2 is clearly a key contribution of this work. It allows images acquired with lenses of different focal lengths to have aligned pixels. However, the description in Section 3.2 is not very clear. The authors should add a diagram to better illustrate the pipeline of the preprocessing.



**Additional Feedback:**

The authors did not provide any code of their benchmarking methods. If possible, the authors are recommended to provide the code they used to evaluate different methods.

**Correctness:**

The claims in the paper seem to be correct. The dataset is constructed properly. The benchmark results are comprehensive and well presented.

**Documentation:**

The documentation of the dataset is sufficient.

**Ethics:**

No personal information can be retrieved from the dataset. I do not see any ethical concerns.

**Relation To Prior Work:**

Prior works of LLE and SR have been properly cited in the introduction and compared in Tables 1 and 2.

**Summary And Contributions:**

In this paper, Aakerberg et al. provide an important dataset that could fill the gap between low-light image enhancement (LLE) and image super-resolution (SR). Unlike existing LLE datasets where the low-light (LL) and normal-light (NL) image pairs share the same spatial resolution, and existing SR datasets where the low-resolution (LR) and high-resolution (HR) image pairs have consistent illumination conditions, the RELLISUR dataset proposed in this work is the first dataset that simultaneously provides paired and aligned LL/LR and NL/HR images of a variety of content. Therefore, RELLISUR, for the first time, enables joint and end-to-end training from LL/LR to NL/HR images. Moreover, the authors benchmark the RELLISUR dataset extensively using 10 different state-of-the-art LLE and SR methods as well as novel joint training approaches, including ESRGAN and MIRNet with an upscaling module. Overall, this paper is very well written and easy to follow. It presents a much-needed dataset that would be highly valuable to the computer vision and image processing community.

---

> ### Author Response · Authors · 2021-09-28
> **Response to reviewer URub**
>
> We thank the reviewer for their feedback and belief in the strengths of our paper. We would like to address some of the concerns of the reviewer below.
>
> Dataset name:
> Thank you for pointing this out. The explanation of RELLISUR was first given in the caption of Figure 1, which we agree is not optimal. We have provided a proper explanation of the name early in the introduction of our revised manuscript.
>
> Auto settings:
> We agree with the unpredictability of the camera's auto settings, which together with changes in the environmental settings can lead to discrepancies, e.g. in brightness levels between the normal-light images, which we also mention as a limitation of our work. However, we have shown that the image pairs in our dataset can be used to train SR, LLE, and joint LLE+SR models which produces high-quality reconstructions, which indicates that the majority of the data is without discrepancies.
>
> Number of sequences:
> Good catch, the total number of normal light LR and HR pairs are of course 2550 and not 2250. We have corrected this mistake in our revised manuscript.
>
> Preprocessing pipeline:
> Thank you for the suggestion. We agree that the pipeline can be difficult to understand without a diagram describing the different steps in the pipeline. We will include such a diagram in our revised manuscript.

---

### Official Review · Reviewer_X1eZ · 2021-09-21
**A Solid Dataset for Low-Light Image Super-Resolution**

**Rating:** 7
**Confidence:** 3
**Correctness:** + Technically correct
**Clarity:** + Clearly written

**Strengths:**

+ This dataset is the first one that provide paired Low-Light Low Resolution (LLLR) and Normal Light High Resolution (NLHR) images. It is useful for developing methods that perform both Low-light enhancement and Super-resolution. The paper shows that the dataset is useful by evaluating state-of-the-art methods on it.

**Weaknesses:**

- It is unclear how well methods trained on this dataset perform on existing datasets such as [6,7,8,9].
- The paper emphasizes on joint Low-light enhancement and Super-resolution but does not explain why the two tasks should be jointly perform. It is not clear what is the benefit of doing them jointly.

**Additional Feedback:**

+ Please see comments above.

====================================

Final comments: I am happy with the answers from authors. I am keeping my original rating.

**Documentation:**

+ Yes

**Ethics:**

+ No concerns.

**Relation To Prior Work:**

+ Well discussed but should include results on other benchmarks as discussed above.

**Summary And Contributions:**

This paper presents the first dataset for training and testing Low-Light Image Super-Resolution networks. The dataset is captured using a carefully designed camera setup and is processed to eliminate artifacts.It contains images of the same scene capture using different focal lengths to attain different resolutions, and the same set of images captured under different exposure settings. The final dataset is provides paired Low-Light Low Resolution (LLLR) and Normal Light High Resolution (NLHR) images. The paper evaluates state-of-the-art methods on this dataset, showing the utility of the dataset.

---

> ### Author Response · Authors · 2021-09-27
> **Response to reviewer X1eZ**
>
> We thank the reviewer for their review and suggestions for improvement. Please find our answers below:
>
>
> Evaluation on traditional SR datasets:
> The datasets you mention (Set5, Set14, BSD100, and DIV2K) are datasets traditionally used in the super-resolution community which is all limited by the fact that they only contain the high-resolution image. This means that the low-resolution image has to be created synthetically (typically with bicubic interpolation), which creates a large domain gap to real low-resolution images which degradation is much more complex than simple downsampling [1]. We mainly include results on dedicated low-light enhancement and super-resolution in our paper to demonstrate that our proposed dataset contains sufficient information to learn these tasks individually, before we move on to the more challenging problem of solving both tasks simultaneously. Since the latter is the main focus of the paper, we believe that not mixing together LLE and SR of real images, with SR of synthetic images will make our paper more understandable.
>
> Joint processing:
> Thank you for pointing out that the benefits of joint processing are not entirely clear in our manuscript. In short, joint processing simply outperforms sequential processing in many cases of image reconstruction. The main reasons for this are the error accumulation occurring in sequential processing, and the possibility that algorithms in the early processing might remove information that could be valuable for subsequent algorithms. This has been proven in previous work dealing with related image reconstruction tasks [1,2,3], and now also in our work for joint LLE and SR. To address your concern, we have clarified the importance of joint processing in our revised manuscript.
>
> [1] A. Lugmayr et al. "NTIRE 2020 Challenge on Real-World Image Super-Resolution: Methods and Results", CVPRW 2020
> [2] Z. Liang, D. Zhang, and J. Shao, “Jointly solving deblurring and super-resolution problems with dual supervised network,” ICME 2019
> [3] T. Klatzer, K. Hammernik, P. Knöbelreiter, and T. Pock, “Learning joint demosaicing and denoising based on sequential energy minimization,” ICCP 2016
> [5] R. Zhou, M. E. Helou, D. Sage, T. Laroche, A. Seitz, and S. Süsstrunk, “W2S: microscopy data with joint denoising and super-resolution for widefield to SIM mapping,” ECCV 2020

---

### Official Review · Reviewer_LzSg · 2021-09-22
**Great Dataset for Both Low-light Enhancement and Image Super-Resolution in Real World.**

**Rating:** 7
**Confidence:** 4
**Correctness:** Nope.
**Clarity:** Yes.

**Strengths:**

1. First large-scale dataset that address the low light image super resolution problem in real world.
2. Benchmark both low light enhancement and super resolution existing methods.

**Weaknesses:**

It's better to show some comparison with existing dataset and results of methods trained on previous dataset to illustrate the importance of the new dataset.

**Additional Feedback:**

Nope.

**Documentation:**

Great details on how to build this dataset. And paper has the link of dataset to check.

**Ethics:**

No.

**Relation To Prior Work:**

Yes.

**Summary And Contributions:**

The paper introduce a novel dataset for both low light enhancement and image super-resolution tasks in real world situation. The authors  manually set different under-exposure level and different focal length to capture low light and low resolution images.  Then they use auto-exposure, auto-white-balance, and auto-focus settings to capture images as ground truth. After that, authors manually filter out low quality images sequence and use Adobe Lightroom and crop to fix lens distortion. With comprehensive analysis and experiments, author prove the dataset can help further research on low light image super resolution task and show a new learning based method is needed to address this task.

---

> ### Author Response · Authors · 2021-09-26
> **Response to reviewer LzSg**
>
> Thank you very much for your review and suggestion for improvement.
>
> We agree that methods trained on existing datasets, where the low-light low-resolution and normal-light high-resolution image pairs would have to be created synthetically, will most likely result in a large performance drop when evaluated on our dataset of real images, since the degradation is much more complex. However, for a fair comparison, we have decided not to include this in our experiments, but we will consider adding a visual example in our revised manuscript to illustrate the difference and to support the importance of our proposed dataset.

---

### Decision · Program_Chairs · 2021-10-10

**Decision:**

Accept

**Comment:**

All reviewers agree that the proposed dataset fills in a gap in the important super-resolution research venue and the paper is well delivered. In addition, the reviewers also confirmed that their concerns are addressed. AC recommends accepting the paper.